# Targeting Pharmacokinetic Drug Resistance in Acute Myeloid Leukemia Cells with CDK4/6 Inhibitors

**DOI:** 10.3390/cancers12061596

**Published:** 2020-06-16

**Authors:** Ales Sorf, Simona Sucha, Anselm Morell, Eva Novotna, Frantisek Staud, Alzbeta Zavrelova, Benjamin Visek, Vladimir Wsol, Martina Ceckova

**Affiliations:** 1Department of Pharmacology and Toxicology, Charles University, Faculty of Pharmacy, Akademika Heyrovskeho 1203, 50005 Hradec Kralove, Czech Republic; sorfal@faf.cuni.cz (A.S.); suchas@faf.cuni.cz (S.S.); staud@faf.cuni.cz (F.S.); 2Department of Biochemical Sciences, Charles University, Faculty of Pharmacy, Akademika Heyrovskeho 1203, 50005 Hradec Kralove, Czech Republic; morellga@faf.cuni.cz (A.M.); novotne7@faf.cuni.cz (E.N.); wsol@faf.cuni.cz (V.W.); 34th Department of Internal Medicine—Hematology, University Hospital Hradec Kralove, Charles University, Sokolska 581, 50005 Hradec Kralove, Czech Republic; alzbeta.zavrelova@fnhk.cz (A.Z.); benjamin.visek@fnhk.cz (B.V.)

**Keywords:** acute myeloid leukemia, CDK4/6 inhibitors, ABC transporters, drug resistance

## Abstract

Pharmacotherapy of acute myeloid leukemia (AML) remains challenging, and the disease has one of the lowest curability rates among hematological malignancies. The therapy outcomes are often compromised by the existence of a resistant AML phenotype associated with overexpression of ABCB1 and ABCG2 transporters. Because AML induction therapy frequently consists of anthracycline-like drugs, their efficiency may also be diminished by drug biotransformation via carbonyl reducing enzymes (CRE). In this study, we investigated the modulatory potential of the CDK4/6 inhibitors abemaciclib, palbociclib, and ribociclib on AML resistance using peripheral blood mononuclear cells (PBMC) isolated from patients with *de novo* diagnosed AML. We first confirmed inhibitory effect of the tested drugs on ABCB1 and ABCG2 in ABC transporter-expressing resistant HL-60 cells while also showing the ability to sensitize the cells to cytotoxic drugs even as no effect on AML-relevant CRE isoforms was observed. All tested CDK4/6 inhibitors elevated mitoxantrone accumulations in CD34^+^ PBMC and enhanced accumulation of mitoxantrone was found with abemaciclib and ribociclib in PBMC of FLT3-ITD^-^ patients. Importantly, the accumulation rate in the presence of CDK4/6 inhibitors positively correlated with *ABCB*1 expression in CD34^+^ patients and led to enhanced apoptosis of PBMC in contrast to CD34^−^ samples. In summary, combination therapy involving CDK4/6 inhibitors could favorably target multidrug resistance, especially when personalized based on CD34^−^ and ABCB1-related markers.

## 1. Introduction

Acute myeloid leukemia (AML) remains a hematological malignity characterized by extremely low curability and survival rate. Despite the overall progress in therapeutic strategies during the past decade [1,2,3], only 28.3% of patients with AML diagnosis currently achieve five-year survival [4]. Clinicians’ decisions regarding optimal treatment strategy are influenced by factors such as age at diagnosis, comorbidities, and risk stratification of AML. Patients eligible for intensive therapy generally undergo induction chemotherapy to reach first remission. Then, that treatment is followed either by courses of consolidation therapy or by bone marrow transplantation to maintain remission status. This strategy, however, is compromised by leukemia stem cells that survive often harsh anticancer treatment and drive drug resistance and relapse [5].

Among other mechanisms, resistance of cancer stem cells can be attributed to such pharmacokinetic phenomena as activity of ATP-binding cassette (ABC) transporters [6,7] or drug inactivation by metabolic enzymes [8]. ABC transporters actively pump a broad range of substrates, including drugs used for AML treatment (e.g., daunorubicin, mitoxantrone), out of cancer cells. They are physiologically expressed in various tissues and biological barriers such as intestine, kidneys, liver, blood-brain barrier, or placenta, where they fulfill a protective role for organism against potentially harmful substances [9]. Since ABC transporters are able to efflux many drugs, they are also well renown perpetrators of drug-drug interactions. Among 49 ABC pumps described to date in humans, ABCB1, ABCG2, and ABCC1 are most dominantly related to resistant AML phenotype [7,10]. Overexpression of ABCB1 and ABCG2 in particular has been confirmed in CD34^+^CD38^−^ leukemia stem cells [11,12], and it has been associated with poor prognosis and outcome of AML therapy [13,14,15]. 

Inactivation of anticancer chemotherapeutics by enzymatic metabolism is another important mechanism of pharmacokinetic drug resistance. Anthracyclines, and especially daunorubicin, remain key therapeutic components in AML induction therapy. This drug, however, undergoes reduction to a 13-hydroxy metabolite via its main biotransformation pathway, thereby leading to the production of the inactive compound daunorubicinol [16,17]. Even as the anticancer activity is lost, this main metabolite is cardiotoxic to the patient [18]. The reduction of anthracyclines, and in part some other drugs such as mitoxantrone [19], is catalyzed by carbonyl reducing enzymes (CREs) from either short-chain dehydrogenase/reductase or the aldo-keto reductase (AKR) superfamily [20,21,22]. Among those CREs reported to play a role in leukemia cells are CBR1, AKR1C3, AKR1A1, AKR1B1, and AKR1B10 [23,24,25,26,27]. Targeting daunorubicin inactivation therefore constitutes a modern pathway that could be exploited in order to prevent and overcome drug resistance [28].

With broadening knowledge of genetic disorders, mutations, and gene deregulations that accompany AML, novel therapeutic approaches have come to the clinical field. CDK4/6 inhibitors (CDKI) are drugs that have successfully provided novel options in the treatment of breast cancer [29]. Two of them, palbociclib and ribociclib, are currently being evaluated in clinical trials for combination therapy of acute leukemias. Importantly, all CDKI approved by Food and Drug Administration to date (palbociclib, ribociclib, abemaciclib) have been found to interact with multiple ABC transporters, and in particular with ABCB1 and ABCG2 [30,31]. The facts mentioned above make them ideal candidates for drug resistance modulation in terms of affecting retention within the cancer cells of such ABC transporter substrates as anthracycline molecules. Although several CDKI drugs have been found to interfere with the activity of CREs [32,33,34], no information has been published to date for palbociclib, ribociclib, and abemaciclib. Knowledge as to how these interactions may affect leukemia cells would therefore be of great therapeutic potential and scientific importance. In this study, we aim to provide information as to whether abemaciclib, palbociclib, and ribociclib interact with ABC efflux transporters and/or CREs, as well as whether such interactions could affect therapeutic outcomes of conventional cytotoxic drugs in peripheral blood mononuclear cells (PBMC) isolated directly from patients with *de novo* diagnosed AML.

## 2. Results

### 2.1. CDKI Enhance Daunorubicin and Mitoxantrone Accumulation in HL-60 Cells

Accumulation assays with daunorubicin and mitoxantrone were conducted in order to determine inhibitory properties of the tested drugs in transporter-expressing resistant leukemia cell lines. All three drugs were able to enhance daunorubicin accumulation in HL-60 ABCB1 cells and mitoxantrone accumulation in HL-60 ABCG2 cells. Ribociclib exhibited similar potency toward ABCB1 and ABCG2, with IC_50_ values of 27.1 and 26.9 µM. The other two drugs were found more effective in ABCB1 inhibition, with the respective inhibitory IC_50_ values of 0.354 µM for abemaciclib and 6.65 µM for palbociclib calculated by considering the effect of model inhibitor as 100% inhibition. Abemaciclib and palbociclib also inhibited ABCG2 with IC_50_s 2.98 µM, and 45.5 µM, respectively, thus exhibiting 8.4-, and 6.8-fold lower potency compared to ABCB1. Details are provided in Figure 1.

### 2.2. CDKI Do Not Inhibit Carbonyl Reducing Enzymes

To identify whether tested drugs inhibit the acute leukemia relevant enzymes involved in anthracycline reduction, a pilot study was conducted using recombinant CBR1, AKR1C3, AKR1A1, AKR1B1, and AKR1B10. The results showed that inhibition by the tested drugs did not reach 50% for any of the enzymes tested even at the 50 µM concentration (Table 1).

In as much as all three tested CDKI revealed some inhibitory effect on recombinant AKR1C3, the inhibition was further verified at cellular level using HCT116 cell line transiently transfected with AKR1C3. The results, however, did not show any changes in intracellular reduction of daunorubicin to daunorubicinol in the presence of abemaciclib, palbociclib, nor ribociclib (Appendix A).

### 2.3. Daunorubicin- and Mitoxantrone-Induced Proapoptotic Behavior of Resistant HL-60 Cells after Exposure to CDKI

To identify whether inhibition of ABCB1 and ABCG2 by the tested drugs leads to apoptotic changes through enhanced accumulation of conventional cytotoxic drugs, annexin V/PI double staining of HL-60 sublines was applied following the treatment with mitoxantrone alone or in combination with the tested drugs at concentrations that were nontoxic but showing inhibition of both ABCB1 and ABCG2, where applicable. When HL-60 control cells were treated by combinations of mitoxantrone with the tested drugs, no shift was observed in comparison to treatment by mitoxantrone alone (Figure 2A). Treatment of HL-60 ABCG2 cells with mitoxantrone and CDKI resulted in significant increase of the annexin V^+^ population from 25.2% when treated by mitoxantrone alone to 56.6% (abemaciclib + mitoxantrone) and 68.6% (ribociclib + mitoxantrone). A nonsignificant proapoptotic pattern (29.7%) was observed, however, with the combination of palbociclib and mitoxantrone (Figure 2B). The single drug treatment with CDKI did not cause increased apoptosis in any of the tested HL-60 cell lines. Interestingly, no proapoptotic effect of the tested drug combinations was determined in HL-60 ABCB1 cells (Figure 2C).

As an additional method to detect late apoptotic changes of HL-60 cells resulting in DNA fragmentation, the sub-G1 fraction was quantified. Neither abemaciclib, palbociclib, nor ribociclib caused a significant proapoptotic effect when applied as a single drug, nor did it significantly affect cell cycle distribution when compared to an untreated control. When added simultaneously with daunorubicin to HL-60 ABCB1 cells, however, all three drugs elevated the sub-G1: from 13.1% to 47.5% for abemaciclib, 30.7% for palbociclib, and 35.6% for ribociclib (Figure 2G). This synergistic phenomenon did not occur in the parental cell line with no ABCB1 expression (Figure 2F). Similarly, after combination of the tested drugs with mitoxantrone in HL-60 ABCG2 cells, the sub-G1 fraction increased from 46.2% to 64.6% for abemaciclib and 64.0% for ribociclib. Palbociclib, on the contrary, did not affect mitoxantrone-induced apoptosis of HL-60 ABCG2 cells (47.0%) (Figure 2E). As expected, no changes were observed in the non-expressing HL-60 control subline (Figure 2D).

### 2.4. CDKI Affect Mitoxantrone Accumulation in CD34^+^ and FLT3-ITD^−^ PBMC

To investigate the effect of the tested drugs on cytotoxic substrate accumulation directly in AML patient cells, the experiments were conducted using 15 AML patient samples, six of which were positive for primitive CD34^+^ blasts. The CDKI were added in concentrations corresponding with maximal reachable plasma levels for standard dosing and were combined with dual ABCB1/ABCG2 substrate mitoxantrone.

Regarding the CD34^−^ population, abemaciclib, palbociclib, and ribociclib did not affect median mitoxantrone accumulation in PBMC compared to untreated control cells (Figure 3A). Taking a detailed look at CD34^+^ cells, the effect of all three drugs was statistically significant, as median mitoxantrone accumulation increased by 37.4%, 33.4%, and 60.5% with abemaciclib, palbociclib, and ribociclib, respectively (Figure 3B). We further divided the patients based on presence of the FLT3-ITD mutation. In the FTL3-ITD^+^ group, no effect of CDKI on mitoxantrone efflux was observed, as the median mitoxantrone accumulation remained on the level of untreated controls (Figure 3D). On the other hand, a significant effect of CDKI was revealed in the FLT3-ITD^−^ population, wherein median mitoxantrone accumulation was increased by 16.4%, 3.58%, and 28.6%, respectively, with abemaciclib, palbociclib, and ribociclib (Figure 3C).

### 2.5. Correlations between Transporter Expression and Effect of CDKI on Mitoxantrone Accumulation in PBMC

We further evaluated the expression of ABC transporters in patient-derived PBMC samples, analyzing whether it is related to the effect of the tested drugs on mitoxantrone intracellular levels. As the transporter inhibition was distinctive in CD34^+^ cells, we first compared *ABCB*1 and *ABCG*2 expression in the CD34^+^ blast to that of other patients. We confirmed significantly increased expression of *ABCB*1 in the CD34^+^ group, with median expression 11.9-fold greater than in the CD34^−^ group (Figure 4A). On the contrary, no differences were observed for *ABCG*2, thus suggesting that functional inhibition in CD34^+^ blasts is being driven mostly by ABCB1, but not ABCG2 activity (Figure 4A). Comparison of expression in FLT3-ITD-mutated patient samples with the FLT3-ITD^−^ group showed that the FLT3-ITD^+^ patients expressed similar levels of *ABCG*2 but 4.88-fold lower levels of *ABCB*1, thus indicating ABCB1 to be the predominant transporter responsible for observed differences in the effects of CDKI on mitoxantrone accumulation between these two groups of patients (Figure 4B).

The predominant role of ABCB1 was further confirmed when we subjected the relationship between the relative expression of *ABCB*1 and *ABCG*2 in PBMC and mitoxantrone accumulation to linear regression analysis. The data show that abemaciclib’s and ribociclib’s functional effect was correlated with *ABCB*1 expression levels. Palbociclib’s inhibitory effect was also increasing with higher *ABCB*1 presence but fell short of statistical significance (*P* = 0.0668) (Figure 4C). On the contrary, no correlation between transporter inhibition and *ABCG*2 expression was found for any of the tested drugs (Figure 4D).

### 2.6. CDKI Enhance Apoptosis of PBMC from CD34^+^ Patients

To determine whether CDKI are able to enhance the proapoptotic effect of mitoxantrone, the annexin V/PI double staining was performed using AML PBMC samples. Because CD34 was determined to be the dominant marker in terms of the tested drugs’ inhibitory effect on ABC transporters, apoptosis of PBMC isolated from patients with CD34^+^ and CD34^−^ phenotypes was compared. We observed no significant changes in PBMC apoptosis after exposure to abemaciclib, palbociclib, or ribociclib when compared to untreated control. When combined with mitoxantrone, none of the tested drugs increased the annexin V^+^ population when compared to single drug treatment by mitoxantrone. On the contrary, the mitoxantrone-induced apoptosis of PBMC derived from CD34^+^ patients was significantly increased by all tested drugs, with median annexin V^+^ populations being enhanced by 5.7–8.2% compared to the sole effect of mitoxantrone (Figure 5A,B). No changes were observed when stratifying the patients based on FLT3-ITD-positivity.

## 3. Discussion

We aimed to examine the effect of the CDKI abemaciclib, palbociclib, and ribociclib on pharmacokinetic mechanisms driving AML multidrug resistance through drug efflux via ABC transporters or its metabolism with CREs.

We first needed to clarify whether the tested drugs inhibit ABC transporters and/or relevant CREs isoforms. Our previous study had confirmed ribociclib as a dual inhibitor of ABCB1 and ABCG2, and Wu et al. had demonstrated a similar effect for abemaciclib [30,31]. We confirmed these results in resistant HL-60 ABCB1 and HL-60 ABCG2, wherein all tested drugs inhibited both transporters, thereby increasing the accumulation of ABCB1 and ABCG2 substrates daunorubicin and mitoxantrone. When evaluating the resulting IC_50_ values in respect to the guidelines provided by the transporter experts and regulatory authorities [35,36], we considered the drugs’ maximum plasma levels achieved after standard dosing in clinical settings [37,38,39]. The inhibitions of both pumps by abemaciclib and ribociclib were of potential clinical significance. Palbociclib’s interaction with ABCG2, meanwhile, resulted in IC_50_ values approximately 100-fold higher than its c_max_, and was therefore outside of the proposed 10-fold range. We took advantage of the inhibitory properties of CDKI to employ annexin V/PI dual staining. In line with the inhibitory data, abemaciclib, ribociclib, but not palbociclib increased mitoxantrone-induced apoptosis of HL-60 ABCG2 cells. Despite clear proapoptotic morphological changes observed in HL-60 ABCB1 cells, we surprisingly observed no annexin V binding even after combination treatment with CDKI and mitoxantrone. We cannot provide a clear explanation for this phenomenon, but we suggest that other drug-induced mechanisms specific for anthracycline-resistant HL-60 cell lines (e.g., altered expression of Bcl-2 and Bcl-xL) might be involved, as reported by Belhoussine et al. [40]. In line with our observation, there are also reports on anthracycline-like drugs indicating that, according to their concentrations and exposure times, they can induce immediate necrosis rather than apoptotic changes [41,42]. Therefore, we further detected the apoptotic and/or necrotic changes using a less-specific method, which is a DNA content/fragmentation-based evaluation of sub-G1 cellular fraction. The data confirmed the hypothesis that abemaciclib and ribociclib sensitize both ABCB1- and ABCG2-expressing cells to mitoxantrone and daunorubicin while palbociclib affected only ABCB1-mediated resistance.

Because daunorubicin remains the key drug component in conventional AML therapy, its inactivation by CREs constitutes another mechanism that might be responsible for diminished intracellular drug concentrations and lead to cellular resistance. We therefore screened the effect of CDKI on recombinant CREs isoforms that previously had been suggested to be related to cellular resistance to AML [23,24,25,26,27,28]. None of the tested drugs, however, achieved 50% inhibition in the applied 50 µM concentration. While AKR1C3 isoform was slightly affected by all three drugs, we further verified the effect by employing a relevant cellular model, confirming no significant inhibition of this isozyme. We can therefore conclude that CDKI do not affect the major metabolic pathway of daunorubicin and are not able to prevent its conversion to the inactive and potentially more harmful metabolite daunorubicinol [18].

Based on the in vitro data, we decided to investigate the modulation of ABC transporters by CDKI ex vivo using PBMC isolated from *de novo* diagnosed AML patients. The expression and activity of ABC transporters in AML cells has regularly been associated with CD34 expression [43,44]. Our results are in accordance with previously reported findings, as all three drugs enhanced dual ABCB1 and ABCG2 mitoxantrone substrate accumulation in CD34^+^ AML blasts while not affecting the intracellular levels of mitoxantrone in PBMC obtained from CD34^−^ patients. This effect was also observed for palbociclib, which was shown to be ineffective in ABCG2 inhibition, thus suggesting that pronounced transporter activity in CD34^+^ cells is predominantly driven by ABCB1. We therefore compared expression of both transporters in CD34^+^ and CD34^−^ patients, revealing significantly higher *ABCB*1 levels in CD34^+^ compared to CD34^−^ patients. In addition, similar levels of *ABCG*2 were detected in both CD34^+^ and CD34^−^ groups. The potential ABCB1 predominance among the AML-expressed ABC transporters was further supported by significant correlation between ABCB1 inhibition by CDKI and *ABCB*1 expression, while no such relationship was determined for ABCG2.

FLT3-ITD mutation is one of the important negative prognostic markers contributing to adverse prognosis in patients with AML [2]. We therefore decided to evaluate whether the effect of the tested drugs on mitoxantrone accumulation would differ in patients with or without FLT3-ITD mutation. We found that CDKI did not inhibit the efflux of mitoxantrone in FLT3-ITD^+^ samples. This is further supported by simultaneously lower mRNA *ABCB*1 expression in the FLT3-ITD^+^ population. We are therefore in agreement with Marzac et al. and Boyer et al., who previously proposed that FLT3^+^ status was linked to low CD34 expression as well as low ABCB1 transporter activity [45,46]. Moreover, abemaciclib and ribociclib, but not palbociclib, increased median mitoxantrone accumulation in FLT3-ITD- group, even though the increase was half when compared to CD34+ subgroup. Taking into account that four out of nine FLT3-ITD-patients were CD34-negative, it appears that the effect on mitoxantrone efflux is dominantly linked to CD34 marker. This would be in agreement with the fact that no differences in apoptosis of cells derived from FLT3-ITD-negative patients were observed.

In general, it has been demonstrated that transporter-mediated efflux is linked to poor prognosis and decreased survival of AML patients [45,47,48]. We show here for the first time that all three tested CDKI were able to enhance apoptosis of AML PBMC from CD34^+^ patients via inhibiting ABCB1-mediated efflux.

## 4. Materials and Methods

### 4.1. Chemicals

Abemaciclib was purchased from Selleckchem (Houston, TX, USA), palbociclib and ribociclib from MedChem Express (Monmouth Junction, NJ, USA). Daunorubicin, mitoxantrone, propidium iodide, RNAse A, NADP+, glucose-6-phosphate, and HPLC grade solvents were bought from Sigma-Aldrich (St. Louis, MO, USA). ABCB1 inhibitor LY335979 and daunorubicinol were obtained from Toronto Research Chemicals (North York, ON, Canada), while ABCG2 inhibitor Ko143 was from Enzo Life Sciences (Farmingdale, NY, USA). Opti-MEM was acquired from Gibco BRL Life Technologies (Rockville, MD, USA). All other cell culture media and reagents were purchased from Sigma-Aldrich, Lonza (Walkersville, MD, USA), and PAA Laboratories (Pashing, Austria).

### 4.2. Cell Cultures

HL-60 cells and their ABCB1- and ABCG2-expressing variants were kindly provided by Dr. Balasz Sarkadi (Hungarian Academy of Sciences, Budapest, Hungary) [49,50,51]. All sublines were grown in L-glutamine containing Roswell Park Memorial Institute (RPMI) 1640 medium supplemented with 10% fetal bovine serum (FBS). The human colorectal carcinoma HCT116 cell line was obtained from Sigma-Aldrich and cultured in Dulbecco’s modified Eagle’s medium with 10% FBS. All cell lines (passages 5 to 20) were grown at 37 °C in an atmosphere of 5% CO_2_ using antibiotic-free medium. 

### 4.3. Daunorubicin and Mitoxantrone Accumulation in HL-60 cells

HL-60, HL-60 ABCB1, and HL-60 ABCG2 cells were preincubated in 1 mL of Opti-MEM in a density of 5 × 10^5^ cells/mL in the presence of a range of concentrations of abemaciclib, palbociclib, and ribociclib. LY335979 (1 µM) and Ko143 (1 µM) were used as ABCB1 and ABCG2 model inhibitors. After 15 min, daunorubicin (HL-60 and HL-60 ABCB1) or mitoxantrone (HL-60 and HL-60 ABCG2) was added for 1 h at a final concentration of 1 µM (37 °C, 5% CO_2_). Subsequently, the samples were centrifuged at 150 × *g* for 5 min, washed in 500 µL ice-cold phosphate buffered saline (PBS), then resuspended again in 500 µL ice-cold PBS and promptly analyzed with a BD FACSCanto II flow cytometer (BD Biosciences, San Jose, CA, USA) using excitation/emission wavelengths 488/670 nm for daunorubicin and 633/660 nm for mitoxantrone.

### 4.4. Screening for Inhibitory Activity on Recombinant CREs

Human recombinant carbonyl reducing enzymes CBR1, AKR1A1, AKR1B1, AKR1B10, and AKR1C3 were expressed and isolated by the *Escherichia coli* expression system as described previously [52]. The screening was conducted using 50 µM abemaciclib, palbociclib, and ribociclib while following a previously published protocol [34].

### 4.5. Inhibitory Assay on HCT116 Cells with Transient AKR1C3 Protein Expression

HCT116 cells were seeded (1.25 × 10^5^ cells/well) in 24-well plates and after 24 h in culture transfected with mammalian pCI_AKR1C3 plasmid [53] as described previously [34,53]. The expression of recombinant AKR1C3 was verified by western blotting using rabbit polyclonal anti-AKR1C3 antibody (ab84327, Abcam, Cambridge, UK). Uniformity of transfection was assumed only when <10% difference was observed between experiments regarding the expression of pCI plasmid containing GFP reporter gene.

In inhibition experiments, the medium was harvested and renewed with Dulbecco’s modified Eagle’s medium with 10% FBS containing 1 µM daunorubicin with or without different concentrations (0.1, 0.5, 1, 5, and 10 μM) of abemaciclib, palbociclib, or ribociclib. After 4 h incubation (37 °C, 5% CO_2_), medium was collected, cells were lysed, and resulting samples were handled and analyzed by ultra high-performance liquid chromatography as described previously [34,53].

### 4.6. Annexin V/PI Staining of HL-60 Cells

HL-60, HL-60 ABCB1, and HL-60 ABCG2 cells were resuspended in 300 µL Opti-MEM giving density of 1.5 × 10^4^ and containing either 1 µM abemaciclib, 10 µM palbociclib, or 25 µM ribociclib, concentrations that inhibited studied transporters and simultaneously did not cause morphological changes of HL-60 cells in accumulation experiments. The cell suspension was preincubated for 15 min, and then, 1 µM mitoxantrone was added. Mitoxantrone, a dual substrate of ABCB1 and ABCG2, was chosen as resistance victim drug for the experiment because there are no fluorescence spillovers with propidium iodide (PI), which limit the usage of daunorubicin in this setup. The cells were incubated for 4 h at 37 °C and 5% CO_2_, washed and subsequently resuspended in 1 × annexin binding buffer (ABB, 195 µL) followed by addition of 5 µL of annexin V/fluorescein isothiocyanate (FITC) conjugate. The suspension was incubated in darkness for 10 min. Cells were then washed once in 1 × ABB and resuspended in 400 µL of ABB. Prior to the measurement, 10 µL of PI solution was added and cells were analyzed by BD FACSCanto II flow cytometer using excitation/emission wavelengths 488/530 nm and 488/670 for FITC and PI, respectively.

### 4.7. Assessment of Sub-G1 Fraction of HL-60 Cells

Estimation of fractional DNA content in the late stage of apoptosis was additionally used to clarify the results of annexin V/PI staining. HL-60, HL-60 ABCB1, and HL-60 ABCG2 cells were harvested and preincubated in a density of 1 × 10^5^ cells/mL with Opti-MEM containing abemaciclib, palbociclib, or ribociclib at a final concentration of 1, 10, or 25 µM, respectively. The fluorescent substrates daunorubicin (0.2 µM) or mitoxantrone (1 µM) were then added and the suspensions incubated for 4 h at 37 °C and 5% CO_2_. Afterwards, the cells were resuspended in 2% FBS in PBS and fixed with ice-cold 70% ethanol for at least 30 min at −20 °C. After fixation, the cells were diluted and washed twice with 2% FBS in PBS. Then, 50 µL of RNAse A (100 µg/mL) was applied to ensure that only DNA would be present. Next, 200 µL of PI (50 µg/mL) was added to stain the DNA content of the cells. The samples were incubated for 1 h at 37 °C and 5% CO_2_. Analysis was performed using a SONY SA3800 spectral cell analyzer. The percentage of cells displaying hypodiploid DNA content is presented as a sub-G1 region of a PI histogram.

### 4.8. Isolation of PBMC from AML Patients

Blood samples were collected from patients with de novo diagnosed AML at the 4th Department of Internal Medicine—Hematology, University Hospital Hradec Kralove following written informed consent approved by the University Hospital Research Ethics Committee (No. 2OL7OS7 LLP). Included into the study were 15 patients, the disease stratification based on WHO classification was six, two, three, two, and two patients for AML with recurrent genetic abnormalities, AML not otherwise specified, therapy-related myeloid neoplasms, AML with myelodysplasia-related changes, and N/A (no cytogenetic available), respectively [54]. Presentation according the European LeukemiaNet (ELN) cytogenetic risk stratification [2], French-British-American (FAB) classification, and basic patient data are presented in Table 2.

PBMC were isolated from whole blood using Ficoll-Paque PLUS gradient solution (GE Healthcare, Sweden) according to the manufacturer’s instructions. The mononuclear cells thus obtained were washed twice in 6 ml PBS and either transferred to RPMI with 10% FBS and immediately used for experiments or stored frozen in liquid nitrogen.

### 4.9. Mitoxantrone Accumulation and Apoptosis Detection in Patients’ PBMC

PBMC were centrifuged at 150 × *g* for 5 min and resuspended in RPMI containing 10% FBS. After 30 min recovery, the PBMC were transferred to Opti-MEM with or without addition of 0.5 µM abemaciclib, 0.5 µM palbociclib, or 7.5 µM ribociclib. The stated concentrations correspond with c_max_ plasma levels of the respective drugs [37,38,39]. After 15 min preincubation, mitoxantrone (1 µM) was added and the mixture was incubated for 4 h (37 °C, 5% CO_2_). The cells were then washed in ice-cold PBS and resuspended in 195 µL of 2% FBS in PBS. The anti-human CD34/PE antibody 4H11 (Invitrogen, Carlsbad, CA, USA) was added and incubated on ice for 30 min. This was followed by washing and resuspending in 1 × ABB and staining by annexin V/FITC conjugate as described above. After subsequent staining with PI, the samples were analyzed using the SONY SA3800 spectral cell analyzer.

### 4.10. Quantitative Reverse Transcription–Polymerase Chain Reaction Analysis

Total RNA was isolated from AML PBMC using TRI Reagent^®^ according to the manufacturer’s instructions. RNA purity and integrity were then checked by measuring A260/A280 and A260/A230 ratios and using 1% agarose gel electrophoresis. The RNA was transcribed to cDNA using the Protoscript II RT kit (New England Biolabs, Ipswich, MA, USA) and the expression of *ABCB*1 and *ABCG*2 analyzed by qPCR run using predesigned TaqMan^®^ assays Hs00184500_m1 *hABCB*1 and Hs01053790_m1 *hABCG*2 in the QuantStudio 6 system (Thermo Fisher Scientific, Waltham, MA, USA). The target gene expression was normalized using human hypoxanthine-guanine phosphoribosyltransferase (*hHPRT*1) (Hs02800695_m1) as a reference gene, which had been chosen based on a housekeeping gene stability analysis using the RefFinder tool (OmicX, Rouen, France). The expression relative to a control sample (which was run in all qPCR experiments) was then calculated by the comparative ΔΔCt method [56].

### 4.11. Statistical Analysis

The data were statistically analyzed in GraphPad Prism 8.0.1 (GraphPad Software, Inc., San Diego, CA, USA). One-way *ANOVA* with Dunnett’s post hoc test or Student’s *t*-test was used for results obtained in vitro, and the nonparametric Mann–Whitney test was used to analyze ex vivo experiments. Half-maximal inhibitory concentrations (IC_50_) were calculated using sigmoidal Hill kinetics while showing also the corresponding 95% confidence intervals. The correlation between expression and function of ABCB1 and ABCG2 was determined by linear regression. All differences were considered significant at *p* < 0.05.

## 5. Conclusions

In summary, our data suggest that combination therapy involving CDKI could favorably target multidrug resistance that is in addition to their own antiproliferative activity, and especially when personalized based on CD34 positivity, FLT3-ITD negativity, and other ABCB1 expression-related markers. Such combination might also be beneficial for older patients who are not candidates for intensive treatment and whose prognosis is generally very poor [2]. The combination of standard treatment with resistance modulators could allow for doses to be reduced and thereby improve tolerance, thus providing an opportunity to treat people whose disease would otherwise be unmanageable.

## Figures and Tables

**Figure 1 cancers-12-01596-f001:**
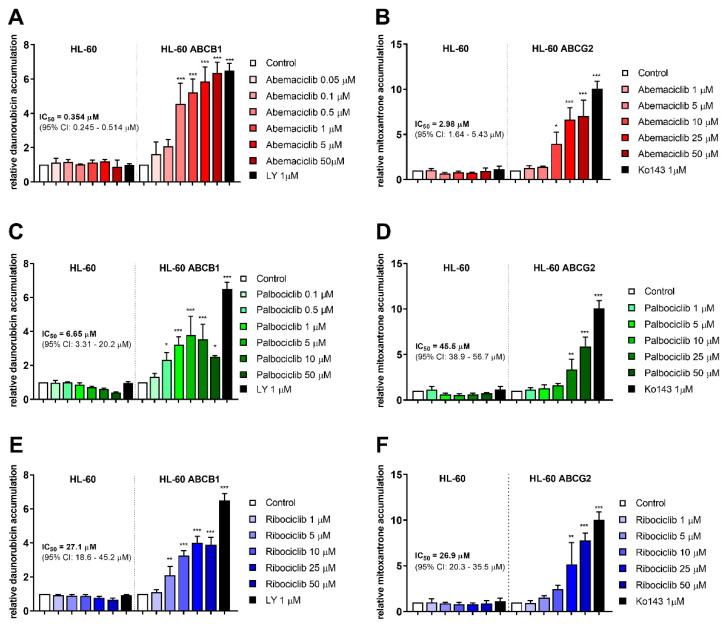
Effect of CDK4/6 inhibitors on daunorubicin and mitoxantrone accumulation in HL-60 cell lines. Increase in accumulation of daunorubicin (**A**,**C**,**E**) and mitoxantrone (**B**,**D**,**F**) was observed as a consequence of treatments by abemaciclib (**A**,**B**), palbociclib (**C**,**D**), and ribociclib (**E**,**F**). Cells were treated with the tested drugs in a range of concentrations or by specific model inhibitors LY335979 (LY), or Ko143 (Ko). After 1 h incubation, fluorescent substrate accumulation was detected and compared to untreated control. Data was analyzed by one-way *ANOVA*; * *p* < 0.05, ** *p* < 0.01, *** *p* < 0.001, *n* ≥ 3.

**Figure 2 cancers-12-01596-f002:**
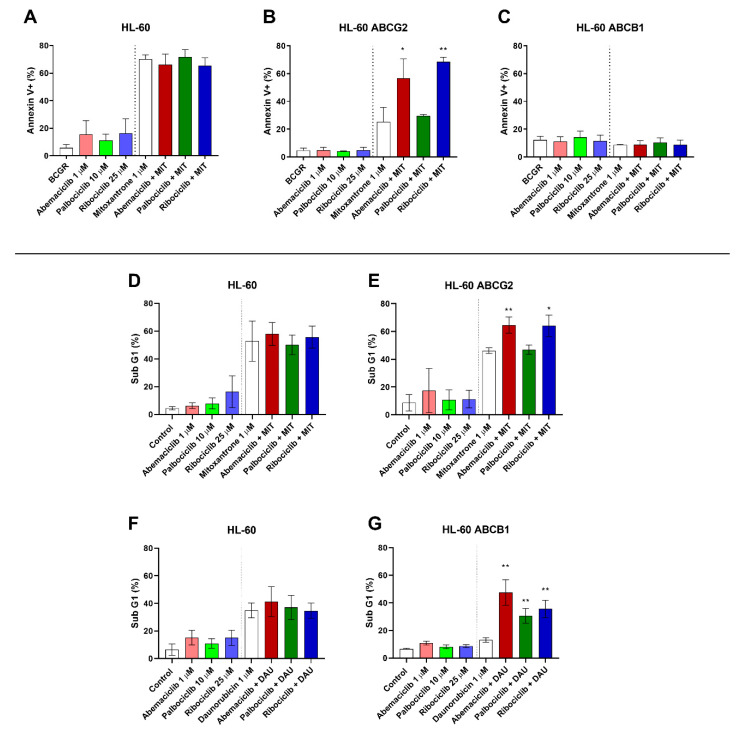
Effect of CDK4/6 inhibitors on mitoxantrone-(MIT) and daunorubicin-(DAU) induced apoptosis of HL-60 sublines. The cells were exposed to the tested drugs, mitoxantrone, and their combinations for 4 h. Subsequently, cells were either stained with fluorescein isothiocyanate (FITC)-labeled annexin V and propidium iodide (**A**–**C**), or fixed with ethanol and stained with propidium iodide (**D**–**G**) and then measured by flow cytometry. Annexin V^+^ populations and sub-G1 fractions were determined in samples treated by the tested drugs with or without MIT or DAU and compared to untreated or MIT- or DAU-treated control by unpaired *t*-test; * *p* < 0.05, ** *p* < 0.01, *n* = 3.

**Figure 3 cancers-12-01596-f003:**
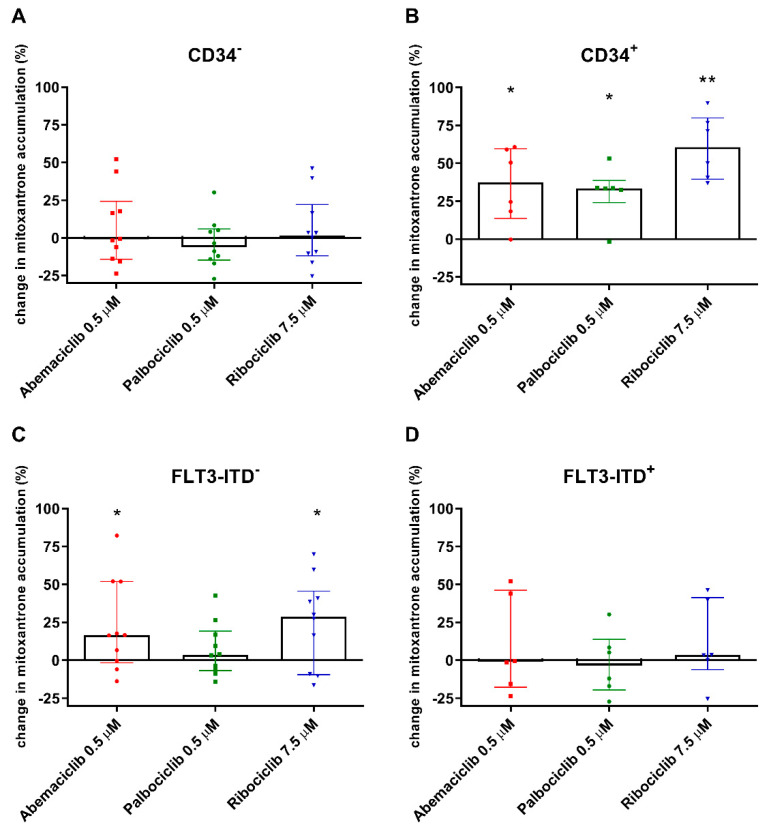
Changes in accumulation of mitoxantrone in peripheral blood mononuclear cells (PBMC) of acute myeloid leukemia (AML) patients exposed to CDK4/6 inhibitors. The accumulation of mitoxantrone (1 µM) was detected in cells with CD34^−^ immunophenotype (**A**) and in CD34^+^ PBMC (**B**) and was compared to untreated control samples of corresponding patients. The accumulation was further evaluated regarding the FLT3-ITD mutation (**C**,**D**). The cells were exposed to mitoxantrone with or without addition of the tested drugs in their c_max_ concentrations for 4 h. Fluorescence was then measured by flow cytometry. The data points and columns represent median ± interquartile range. Statistical analysis was performed by Mann–Whitney test; * *p* < 0.05, ** *p* < 0.01.

**Figure 4 cancers-12-01596-f004:**
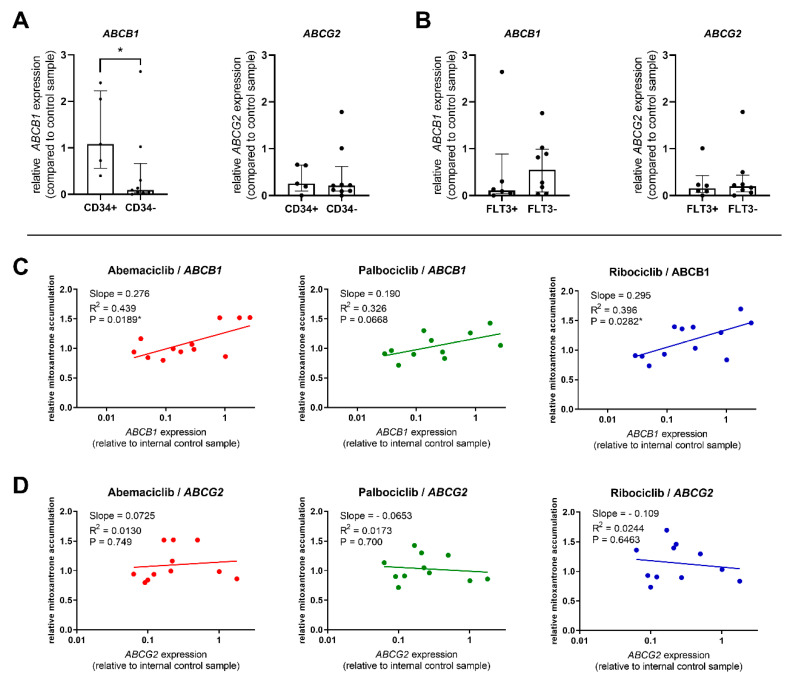
Comparison of *ABCB*1 and *ABCG*2 expression; relation to transporter inhibition. Expression of target genes was compared between CD34^+^ and CD34^−^ patients (**A**) and between FLT3-ITD^+^ and FLT3-ITD^−^ patients (**B**). The expression was normalized using the *HPRT*1 reference gene and is reported as relative to the control sample run in all qPCR plates. The columns represent median ± interquartile range and a comparison was conducted using Mann–Whitney test, * *p* < 0.05. Correlation between mitoxantrone accumulation in PBMC treated by abemaciclib, palbociclib, or ribociclib and the expression of *ABCB*1 (**C**) or *ABCG*2 (**D**) was further calculated. Data were analyzed by linear regression, and correlation was considered significant at *p* < 0.05.

**Figure 5 cancers-12-01596-f005:**
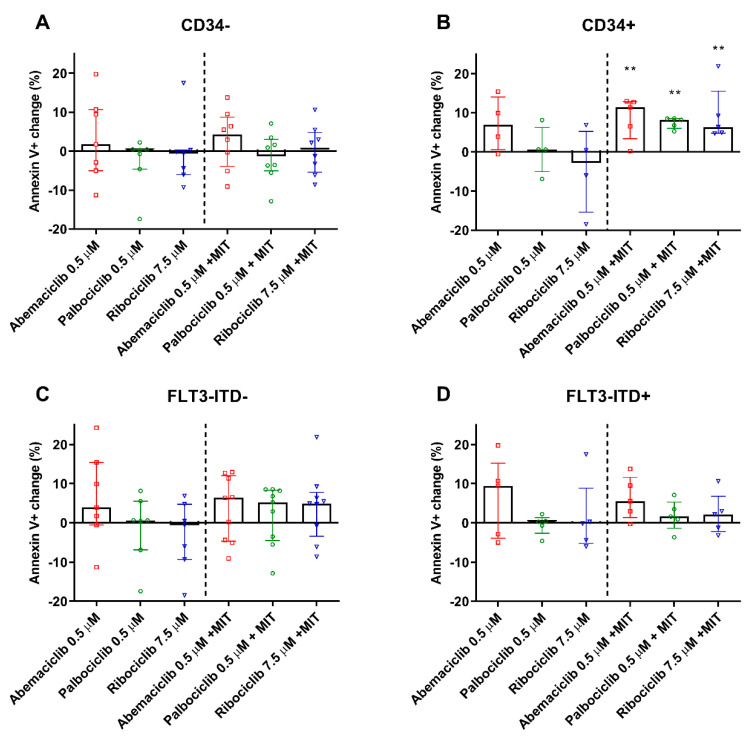
Apoptosis detection in PBMC of AML patients. Percentage change in apoptotic PBMC of CD34^−^ (**A**), CD34^+^ (**B**), FLT3-ITD^−^ (**C**), and FLT3-ITD^+^ (**D**) patients treated by the tested drugs and compared to untreated (left side) or mitoxantrone-(MIT) treated (right side) control samples of the respective patients. The cells were exposed to tested drugs for 4 h and subsequently stained by annexin V/FITC and propidium iodide. Fluorescence was then detected by flow cytometry. The columns represent medians ± interquartile range. Data was analyzed by Mann–Whitney test; ** *p* < 0.01.

**Table 1 cancers-12-01596-t001:** Inhibition of recombinant CRE (%) by abemaciclib, palbociclib, and ribociclib. Each value represents the mean ± SD from three independent experiments.

Inhibition (%) (50 μM Inhibitor Concentration)					
	CBR1	AKR1A1	AKR1B1	AKR1B10	AKR1C3
Abemaciclib	0	0	0	2.08 ± 0.40	26.34 ± 1.30
Palbociclib	0	0	5.54 ± 3.40	0	23.51 ± 6.67
Ribociclib	0	0	4.58 ± 2.41	7.41 ± 1.08	36.18 ± 2.83

**Table 2 cancers-12-01596-t002:** Basic characteristics of patients included in the study according to French-British-American (FAB) classification and European LeukemiaNet (ELN) risk stratification [2,55].

Sex	Age	FAB	Karyotype	CD34	Mutations	ELN Risk
M	61	M4	46,XY	−	*NPM*1*A*, *IDH*2	favorable
F	64	M4	-	−	*FLT*3, *NPM*1*A*	N/A
F	64	M4	46,XX	+	*FLT*3, *NPM*1*D*	favorable
M	67	M2	44,XY,del(2)(q?21q?31), del(5)(q12q34),del(6)(q21q25),add(8)(q24),der(9) t(?8;9),dic(16)t(16;17) (?;q10),del(20q) [25]	+	*TP*53	adverse
M	69	M5a	46,XY	−	*NPM*1*A*	favorable
M	71	M2	−	+	−	N/A
M	54	M4	46,XY	−	*FLT*3, *NPM*1*A*	intermediate
M	65	M1	46,XY	+	*AML*1-ETO	favorable
F	74	M2	−	−	*FLT*3, *NPM*1*A*	favorable
M	63	M2	46,XY,t(1;3)(p36;q21), der(1)t(1;3)(p36;q21) [25]	−	−	intermediate
M	54	M1	46,XY	−	*FLT*3	intermediate
F	66	M2	46,XX	−	*FLT*3, *NPM*1*D*	intermediate
F	68	M2	44,XX,-2,del(5)(q21q34), der(13)t(2;13;?),?del(17p)-18, del(20)(q12) [30]	+	*TP*53	adverse
F	70	M4	46,XX	+	−	intermediate
M	30	M2	46,XY	−	*NPM*1*A*	adverse

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
