# Peer review of "Targeting Pharmacokinetic Drug Resistance in Acute Myeloid Leukemia Cells with CDK4/6 Inhibitors"

_cancers, 2020, doi:10.3390/cancers12061596_

Round 1

Reviewer 1 Report

I have reviewed the manuscript. This is a nicely written paper with some interesting and encouraging results. I would suggest some minor changes.

1) I guess the authors did not include APL cases. I think this should be stated. An do those compounds have an effect on APL at all? This could be interesting to be stated, although I guess no APL cases nor cell lines were included.

2) How did the authors decide the doses of abemaciclib, palbociclib, and ribociclib used in the experiments. This should be stated and maybe discussed.

3) The revision should be submitted with the correct sequencing of the manuscript parts. In this version introduction is followed by results, and then discussion, methods and conclusion. These should be in the correct order.

Reviewer 2 Report

In the manuscript “Targeting pharmacokinetic drug resistance in acute myeloid leukemia cells with CDK4/6 inhibitors” authors evaluate combination therapy involving CDK4/6 inhibitors as favorably target multidrug resistance. Overall the aim is interesting there are some points to be discussed:
Minor revisions
1)    In Figure 1 (pg 3/19) Ribociclib presented the same concentration range between figure 1E and F experiments, while the other molecules have different concentration ranges. Uniform the concentration used.
2)    The results obtained in figure 2C (pg 4/19) can be given as supplementary data.
3)    In paragraph 2.3 apoptotic results were confirmed in terms of sub-G1 fraction quantification; please provide the same results provided in terms of annexin V/PI double staining (combination with Daunorubicin is lacking).
4)    Analyzing sub-G1 fraction, did you observed c Analyzing sub-G1 fraction, did you observe cell cycle alterations? Please in case of positive response provide cell cycle data. 
5) Results obtained on PBMCs from AML patients has to be related to FAB classification in order to correlate the results obteined in HL-60 cell line and APL patients; A table with patients characteristics has to be provided.
Major revisions
1)    The authors underlined as anthracyclines, especially daunorubicin, remain today key therapeutic components in AML induction therapy and as the reduction of anthracyclines is catalyzed by carbonyl reducing enzymes (CREs). On the contrary the selected cell line model derive from Acute Promyelocytic Leukemia, currently treated with trans-retinoic acid (ATRA), and so probably it wasn’t the best model for the current study.

Reviewer 3 Report

This is an interesting paper on the use of CDKI as an enhancer of sensitivity to mitoxantrone or daunorubicin studied in a single cell line model and 12 primary PBMC with isolation for CD34+ and CD34- components.

Background: The role of ABCB1 and ABCG2 is extremely important and is very brief. These should be expanded and mention of the baseline of these transporter in normal tissue should be introduced.

 Where else are the transporters expressed and how many this impact toxicity profile?

What happens in the normal CD34+ cell?

Figure 1 legend is missing.

What are the trial concentrations for the drugs in comparison to the doses used here?

Is apoptosis the correct measure? maybe differentiation occurs and then leads to cell death. Can CD11c be measured or May grunwald geimsa staining of the samples.

Figure 2: What is the the relationship between the transporters and the enzymatic inhibition. This seems to come from out of no where and as a result seems out of place and confusing to the reader. What is the positive control for this experiment? and why use a colorectal cancer line in this case? Are there no AML models that can be used? the legend also does not indicate what C) is.

Figure 3: How was the red color of the daunorubicin accounted for in the flow assays. Is this the reason to focus on the mitoxantrone in the latter experiments?

Figure 4: why is palbociclib response different between the CD34+ and FLT3- - if this is a class effect.

figure 5: gene expression by PCR is measured by Delta delta CT values, this should be used also was more than one control used? How do we know that the control do not change.

Figure 6: where is the FLT 3 data?

General comments:A table of the patient data may be easier to follow.

Was there any primary FLT3 data?

Round 2

Reviewer 3 Report

The challenges with the color spectrum should be mentioned in the methods.

The figure legends are confusing and missing for figure 2 and then all the numbers are off for the legend makes the read very confusing.

The cytogenetics should  be listed in the table for each patient.
